# Roles of MicroRNAs in Osteogenesis or Adipogenesis Differentiation of Bone Marrow Stromal Progenitor Cells

**DOI:** 10.3390/ijms22137210

**Published:** 2021-07-05

**Authors:** Ya-Li Zhang, Liang Liu, Yaser Peymanfar, Paul Anderson, Cory J. Xian

**Affiliations:** UniSA Clinical and Health Sciences, University of South Australia, Adelaide, SA 5000, Australia; yali.zhang@mymail.unisa.edu.au (Y.-L.Z.); liang.liu@csl.com.au (L.L.); yaser.peymanfar@mymail.unisa.edu.au (Y.P.); Paul.Anderson@unisa.edu.au (P.A.)

**Keywords:** BMSC differentiation, bone/fat formation, microRNAs

## Abstract

Bone marrow stromal cells (BMSCs) are multipotent cells which can differentiate into chondrocytes, osteoblasts, and fat cells. Under pathological stress, reduced bone formation in favour of fat formation in the bone marrow has been observed through a switch in the differentiation of BMSCs. The bone/fat switch causes bone growth defects and disordered bone metabolism in bone marrow, for which the mechanisms remain unclear, and treatments are lacking. Studies suggest that small non-coding RNAs (microRNAs) could participate in regulating BMSC differentiation by disrupting the post-transcription of target genes, leading to bone/fat formation changes. This review presents an emerging concept of microRNA regulation in the bone/fat formation switch in bone marrow, the evidence for which is assembled mainly from in vivo and in vitro human or animal models. Characterization of changes to microRNAs reveals novel networks that mediate signalling and factors in regulating bone/fat switch and homeostasis. Recent advances in our understanding of microRNAs in their control in BMSC differentiation have provided valuable insights into underlying mechanisms and may have significant potential in development of new therapeutics.

## 1. Introduction

Bone marrow mesenchymal stem cells (BMSCs) have the potential and capacity to differentiate into multiple cell types, such as chondrocytes, osteoblasts, and adipocytes [1,2]. The commitment and differentiation of BMSCs are tightly controlled by factors of their micro-environment, as changed signals and molecules can lead to their abnormal differentiation causing changes in volumes of trabecular bone and bone marrow fat. Bone loss, which is replaced with increased bone marrow fat, occurs under pathological conditions, including aging, menopause, chronic glucocorticoid treatment, or cancer chemotherapy treatment [3,4]. This phenomenon has been shown to occur due to a shift in BMSC differentiation to favour the formation of fat cells or adipocytes (adipogenesis) over bone-forming cells or osteoblasts (osteogenesis). Under these conditions, there are some similarities in the pathological stresses characterized by suppressed osteogenic signals and elevated adipogenic signals, leading to reduced osteogenesis and enhanced adipogenesis differentiation. This change is associated with the development of osteoporosis and increased fracture risk [5,6]. While numerous studies have described the cellular mechanisms of bone/fat balance in the bone marrow [4,7,8], the underlying mechanisms for unbalanced bone/fat formation from the BMSCs has not been fully established.

Osteogenesis or adipogenesis arises due to BMSC differentiation responding to activation of specific signalling pathways. Many recent or current investigations have been trying to identify key factors that can control the activation or attenuation of major signalling pathways and transcription factors in BMSCs. MicroRNAs are 19–25 non-coding RNA molecules that play essential functions in RNA silencing and post-transcriptional regulation of encoding genes by interacting with the 3’ untranslated region (3’ UTR) of the mRNA transcripts that present similar amino acid sequences to inhibit or degrade the target genes [9,10]. MicroRNAs are critical mediators of many biological processes, including tissue formation, cell differentiation, and cancer development [11,12]. Currently, knowledge is continuously accumulating related to microRNA involvement in BMSC differentiation regulation [13,14,15]. By conducting microRNA array and profiling/pathway analyses, certain microRNAs and their specific targets have been identified to control the process of BMSC differentiation, which has led to the elucidation of the functions and regulations of some microRNAs in regulating bone/fat formation [16,17].

In this review, we have first summarised the key signalling pathways and molecular factors associated with bone/fat formation and how microRNAs are involved and regulating these signalling pathways and factors. In addition, recent developments, and exploration of using microRNAs in early diagnostics of bone defects, preventing bone loss, and bone marrow adiposity are briefly discussed.

## 2. Key Signalling Pathways Involved in Bone/Fat Formation

Two key signalling pathways have so far been identified to control bone or fat formation: namely, the Wnt/β-catenin and the TGF-β/BMP/Smad signalling pathways.

### 2.1. Wnt/β-Catenin Signalling

The first observation indicating the link between bone biology and Wingless (Wnt)/β-catenin signalling was made more than ten years ago [18]. Yavropoulou and Yovos studied Wnt signalling mutants and found that these mutants lead to excessive bone formation [19]. Additionally, Wnt/β-catenin has been identified as an essential regulatory signalling pathway for adipogenesis [20,21,22]. Activated Wnt/β-catenin signalling stimulates osteogenic gene expression, including runt-related transcription factor 2 (RUNX2), distal-less homeobox 5 (DLX5), and osterix (OSX) [23,24]. However, inhibition of Wnt/β-catenin signalling induces adipogenic differentiation via increasing the expression of adipogenic-related genes, such as peroxisome proliferator-activated receptor γ (PPARγ) and CCAAT/enhancer-binding protein α (C/EBPα) [25,26]. Further studies have indicated that β-catenin interacts with PPARγ resulting in β-catenin degradation, which attenuates Wnt/β-catenin signalling and bone formation [27]. Therefore, Wnt/β-catenin signalling is well-characterized and strongly implicated in bone/fat regulation (Figure 1).

Recent investigations into Wnt/β-catenin signalling have found that at least 19 Wnt ligands (e.g., Wnt1, 2, 3a, 3b, 4, 8, and 10b) can activate this pathway [28]. Activation of Wnt/β-catenin signalling starts with the Wnt ligands binding to a receptor complex that is composed of specific frizzled (FZD) seven-transmembrane-span protein and low-density lipoprotein receptor-related protein 5/6 (LRP-5/6) [12]. Intracellular disheveled (DVL) is then activated, disrupting the inhibitory effects on β-catenin, leading to translocation of β-catenin into the nucleus [29,30]. Intranuclear β-catenin then forms a transcriptional complex with the members of T-cell factor (TCF)/lymphoid enhancer-binding factor (LEF) to together mediate target gene expression [29]. In the absence of Wnt ligands, β-catenin is inhibited by a degradation complex, including glycogen synthase kinase 3β (GSK-3β), axin, and adenomatous polyposis coli (APC) [31]. In addition, the Wnt signalling pathway is negatively regulated by secreted Wnt antagonists. Secreted frizzled-related protein 1 (sFRP-1) and Wnt inhibitory factor 1 (WIF-1) inhibit the interactions between Wnts and the FZD receptor [32]. Sclerostin and Dickkopf (Dkk) families, including Dkk-1, -2, and -4, bind to LRP-5/6, inhibiting LRP-5/6 co-receptor activity [33].

### 2.2. TGF-β/BMP/Smad Signalling

Transforming growth factor β (TGF-β)/Smad signalling favours early differentiation of osteoprogenitors by promoting DLX5 expression [34,35,36]. DLX5 then induces the expression of RUNX2 and OSX [37]. However, TGF-β/Smad signalling is a double-edged sword in bone homeostasis. Active TGF-β/Smad signalling negatively contributes to osteoblast maturation, mineralization, and osteocyte transition [36].

Bone morphogenetic proteins (BMPs), as members of TGF-β superfamily, also have a strong modulatory effect on osteogenesis/adipogenesis differentiation. Based on the homology of amino acid sequences, bone-inducing BMPs have been divided into four subgroups: BMP2/4, BMP5/6/7, BMP9/10, and BMP12/13/14 subgroups [38]. Other BMPs have not shown osteogenic features but act differentially. For example, BMP3 is an antagonist of BMP2/4 signalling [39]. While BMP/Smad signalling enhances each step of osteogenesis and differentiation into osteocytes [36], BMP/Smad signalling is also known to promote adipogenesis by increasing the expression of PPARγ, a master transcription factor of fat formation [40]. Unlike BMP/Smad signalling, TGF-β/Smad signalling has shown inhibitory effects on adipogenesis and marrow fat formation [36]. Therefore, TGF-β or BMP signalling has multiple roles in regulating bone/fat homeostasis (Figure 2).

TGF-β/BMP ligands, once bound to TGF-β receptor (TGF-βR) I/II or BMP receptor (BMPR) I/II complex, can trigger Smad-dependent signalling pathways. Three categories of Smads are involved in TGF-β/BMPs signal transduction, containing receptor-Smads (R-Smads), common-Smads (C-Smads), and inhibitory-Smads (I-Smads) [41]. R-Smads, including Smads 1, 5, and 8 for the BMPs signalling pathway and Smads2 and 3 for the TGF-β signalling pathway, are activated through phosphorylation. Upon activation, R-Smads release from the receptor complex and form a complex with C-Smad called Smad4 [42]. The complex R-Smads/C-Smad is translocated into the nucleus and modulates the gene expression of diverse transcriptional factors. I-Smads negatively regulate the TGF-β/BMPs signalling pathway. For example, Smad6 can competitively bind to R-Smads, preventing the R-Smads/C-Smad complex formation [43]. Smad7 interacts with E3 ubiquitin ligase proteins, such as Smad ubiquitin regulatory factor 1 and 2 (SMURF 1/2), resulting in degradation of R-Smads [43].

## 3. Key Factors and Markers Involved in Bone or Fat Regulation

### 3.1. Key Factors and Markers Involved in Osteogenesis and Bone Formation

Bone formation is a highly restricted process associated with the commitment of numerous BMSCs towards osteoprogenitor cells, followed by the formation of pre-osteoblasts and mature osteoblasts. Many studies have been conducted to understand the factors regulating gene expression during BMSC commitment, osteoblast differentiation, and maturation.

BMPs, members of TGF-β and osteoblast-specific transcription factor OSX are considered important factors to control the commitment of MSCs to the osteoblast lineage [44]. RUNX2 is required for the proliferation of osteoprogenitors and is considered the earliest transcription factor to determine the osteoblast lineage [45,46]. Once osteoprogenitor cells begin to differentiate into pre-osteoblasts, there is a proliferation phase, marked by the high expression of alkaline phosphatase (ALP), a vital enzyme to initiate and regulate bone mineralization [47]. The expression level of ALP is upregulated depending on osteoblast maturation, and thus ALP is considered as a specific molecular marker for the early stage of osteoblast differentiation [48]. In addition, other transcription factors, such as activating transcription factor 4 (ATF4) and DLX5, are all involved in osteoblast differentiation.

The transition of pre-osteoblasts to mature osteoblasts is marked by the secretion of molecules such as osteocalcin (OCN), bone sialoprotein (BSP), osteopontin (OPN), and type 1 collagen, which are essential for osteoid formation and mineralization [49]. OCN is thought to be related to mature osteoblast activity and, therefore, is seen as a marker for late-stage osteoblastogenesis [50]. Type 1 collagen forms osteoid, which is the matrix upon which osteoblasts stimulate the precipitation of calcium salts and phosphorous within the newly formed osteoid to form bone mineral, or hydroxyapatite [51].

### 3.2. Key Factors and Markers Involved in Adipogenesis and Fat Formation

Adipocyte differentiation is characterized by changes of expression of a series of specific genes and factors that determine the adipocyte phenotype. These changes in gene expression primarily happen at transcriptional levels, which are reflected at different stages of adipogenesis. Numerous studies have explored the molecular mediators that drive the transition from BMSCs to mature adipocytes. The development of adipocytes can be briefly described into two phases [52,53]. The first phase is the commitment of BMSCs to pre-adipocytes, a process influenced by various extracellular factors, including BMPs, insulin/insulin-like growth factor-1 (IGF-1) and TGF-β [54]. The presence of TGF-β negatively regulates adipogenesis and suppression of adipogenesis might be obtained through TGF-β supplementation [54]. While the morphological difference between precursor cells and pre-adipocytes cannot be identified, pre-adipocytes lose the differentiation potentials into other cell types [54]. The second is known as terminal differentiation, in which the pre-adipocytes obtain the characteristics of mature adipocytes and gradually obtain their physiological functions, such as lipid deposition and transport and secretion of adipocytic factors [55].

Two transcription factors, C/EBPs and PPARγ, are recognized to be involved in pre-adipocyte growth arrest and the promotion of terminal adipocyte differentiation and accumulation of lipid droplets [56,57,58]. Some family members of C/EBP, including C/EBPα, C/EBPβ, C/EBPγ, and C/EBPδ, have been recognized to be expressed in adipocytes. The early induction of C/EBPβ and C/EBPδ stimulates the induction of C/EBPα and PPARγ [52,59]. Activated C/EBPα and PPARγ work cooperatively in adipogenesis by activating other adipogenic-specific genes and lipogenic enzymes. C/EBPγ likely inhibits adipocyte differentiation by inactivation of, and heterodimerization with, C/EBPβ [52,60]. Furthermore, activated PPARγ can promote pre-adipocytes to exit from the cell cycle, triggering increased energy delivery to cells [56,61]. C/EBPα and PPARγ are well characterized as primary transcription factors in adipogenesis and fat formation. Other factors are involved in regulating adipogenesis by modulating C/EBPα and PPARγ expression. Sirtuin I (SIRTI), a histone deacetylase that affects energy homeostasis, inhibits adipogenesis by binding to PPARγ. In SIRTI adipocyte-specific knockout mice, PPARγ was found to be hyperacetylated, which lead to a decrease in PPARγ phosphorylation and increased insulin sensitivity [62].

## 4. MicroRNA Biogenesis

MicroRNAs are small non-coding RNA molecules that regulate the post-transcriptional gene expression by inhibiting target mRNA translations or promoting transcript degradation [9,10]. It has been estimated that in the human genome, there are more than 1400 mammalian microRNAs and 45,000 target sites, which cover 60% of genes [63]. Unlike proteins where translation occurs from coding regions of the genome, microRNAs arise from intronic, intergene, and polycistronic clustered regions [64].

During the process of microRNA biogenesis, the primary microRNA strand (pri-microRNA) forms multiple stem-loop structures in the nucleus, which are canonically processed to mature microRNAs. Briefly, the pri-microRNA transcripts undergo trimming by the Drosha-DGCR8 RNase complex and proceed into pre-microRNAs. Upon transfer to the cytoplasm via exportin-5/GTP61, the pre-microRNAs are transformed into microRNA duplexes (guide and passenger strands) after being further cleaved by a Dicer complex containing Dicer, argonaute 2 (AGO2), and transactivation response element RNA-binding protein (TRBP) [64].

During microRNA-involved gene inhibition, the guide strand is often loaded into a microRNA-inducing slicing complex, composing of AGO2 and GW182 [65]. The microRNA-inducing slicing complex can bind to the 3’ UTR of target mRNA transcripts [65]. Subsequently, the target mRNA will be subjected to translation suppression or degradation, as shown in Figure 3.

## 5. MicroRNAs Involved in Bone/Fat Formation

MicroRNAs have shown hormone-like activities. Once secreted by host cells into extracellular fluids, microRNAs are either transported by forming a complex with AGO2 or by vesicles, such as exosomes [66]. Therefore, similar to hormones, microRNAs might have autocrine, paracrine or endocrine regulatory functions [67]. Recent studies have shown that osteoblast differentiation from BMSCs involves regulation from autocrine and paracrine microRNA signalling [68]. Furthermore, Fischer et al. found the miR-327-based autocrine regulatory network that mediates adipocyte differentiation [69]. Thus, notably, microRNAs may interact with recipient cells for bone/fat formation regulation through multiple mechanisms and many factors.

The currently known microRNAs and their potential targets related to osteoblast and adipocyte differentiation are detailed in Table 1. From this list, it can be seen that microRNAs and their targets are mainly involved in the two key signalling pathways identified to control bone or fat formation: namely, the Wnt/β-catenin and the TGF-β/BMP/Smad signalling pathways.

### 5.1. MicroRNAs Involved in Osteogenesis and Bone Formation

#### 5.1.1. MicroRNAs Involved in Wnt/β-Catenin Signalling

Dickkopf-1 (DKK1), an antagonist of the Wnt/β-catenin signalling pathway, is recognized as a critical biomarker for osteoporosis [114]. The expression of DKK1 is highly and adversely correlated with bone formation. Preventing or treating bone loss might be obtained by blocking endogenous DKK1. Previous work found miR-433-3p targeting DKK1 expression, and in rat bone marrow-derived osteoblasts transfected with lentivirus vector with microRNA-433-3p construct, the levels of ALP and mineral deposition were significantly increased, indicating the positive role of microRNA-433-3p in osteoblast differentiation [70]. MicroRNA-107 was also found to regulate the Wnt/β-catenin signalling pathway by targeting DKK1 in vitro [71]. In addition, microRNA-335-3p was found to regulate DKK1 protein levels in osteoblasts [72]. In vivo studies showed that microRNA-355-3p significantly decreased DKK1 protein levels in bones, and C3H10T-1/2 and MC3T3-E1 cells transfected with microRNA-355-3p were consistently found to have a dramatic increase in the phosphorylation of glycogen synthase kinase 3β (GSK-3β), which activates the Wnt/β-catenin signalling pathway and promotes the process of bone formation [72]. Recently a study showed that by targeting DKK1, overexpression of microRNA-217 promotes the nuclear translocation of β-catenin and further induces BMSC differentiation for osteoblasts [73]. Furthermore, for microRNA-29, which is known to be necessary for human osteoblast differentiation, transfection with microRNA-29 mimic reduces the expression of DKK1, leading to positive regulation of Wnt/β-catenin signalling and osteoblast differentiation [74].

MicroRNA-542-3p has been demonstrated to modulate osteogenesis positively by targeting secreted frizzled-related protein 1 (sFRP-1), a negative regulator of the Wnt/β-catenin signalling pathway [75]. Significant changes in microRNA-542-3p and sFRP-1 expression were observed in an ovariectomized rat model, and there is a negative correlation between microRNA-542-3p and sFRP-1 expression levels [75]. MicroRNA-542-3p transfection in BMSCs increased the expression of osteoblast-specific makers, such as ALP, RUNX2, and OCN, suggesting that microRNA-542-3p promotes BMSC differentiation to osteoblasts [75]. In addition, in vivo studies showed that inhibition of microRNA-542-3p reduced bone formation, confirming that microRNA-542-3p plays a critical role in osteoblast differentiation and bone formation by inhibiting sFRP-1 expression [75]. Furthermore, microRNA-218 was reported to promote osteoblast differentiation, and studies have shown that microRNA-218 downregulates sclerostin, DKK2, and sFRP-2, stimulating Wnt/β-catenin signalling. That activated Wnt/β-catenin signalling, in turn, upregulates microRNA-218, thereby enhancing Wnt activity. This positive loop drives osteoblast differentiation [77].

Several microRNAs can target the 3’ UTR of GSK-3β or APC, which positively controls the Wnt/β-catenin signalling pathway. Overexpression of microRNA-346 significantly induced osteogenesis, while the deletion of microRNA-346 repressed osteoblast differentiation [78]. Further results indicated that microRNA-346 overexpression activated the Wnt/β-catenin signalling pathway [78]. It was also reported that microRNA-26a targets GSK-3β to activate Wnt/β-catenin signalling to promote the osteogenic differentiation of BMSCs [79]. Consistently, co-implantation of biomaterial β-tricalcium phosphate and BMSCs transfected with microRNA-26a in a mouse model of skull defects exhibited increased bone regeneration and bone formation [115]. Zhao et al. found that microRNA-199b-5p is upregulated during osteogenesis in human BMSCs (hBMSCs) [80], and inhibition of microRNA-199b-5p notably reduced osteogenesis by directly targeting GSK-3β [80]. Furthermore, microRNA-27 and microRNA-142-3p were identified as positive molecules for osteoblast differentiation using human fetal osteoblastic 1.19 (hFOB1.19) cells [81,82]. In vitro results showed that microRNA-27 and microRNA-142-3p directly targeted APC and inhibited gene expression of APC, and that the expression of microRNA-27 and microRNA-142-3p is positively correlated with nuclear translocation of β-catenin, resulting in increased osteoblast differentiation [81,82].

Some microRNAs serve as negative regulators by silencing the Wnt signalling receptors and targeting Wnt ligands. An osteogenic inhibitor, microRNA-23a, was significantly downregulated during osteogenesis of human BMSCs (hBMSCs), inhibiting this process by targeting the LPR-5 receptor and thereby causing attenuated Wnt/β-catenin signalling [89]. Similarly, microRNA-30e was identified as a negative regulator for bone formation. It has been reported that overexpressed microRNA-30e inhibits osteoblast formation from mouse BMSCs by targeting LRP-6 directly [90], as silenced LRP-6 negatively regulates osteogenesis by blocking the Wnt ligand and co-receptor binding, affecting Wnt/β-catenin signalling [90]. MicroRNA-127 and microRNA-136 were found to be upregulated in ovariectomized mouse osteoporosis models. Transfection of microRNA-127 inhibitor enhances osteogenesis in UAM-32 cells as measured by ALP activity and mRNA expression of RUNX2, and transfection of microRNA-136 precursor has inhibitory effects on osteoblastic differentiation, indicating that both microRNA-127 and microRNA-136 negatively regulate osteoblast differentiation [91]. Feng et al. found abnormal bone tissues and impaired bone quality in microRNA-378 transgenic mice [92]. It was observed that microRNA-378 suppresses osteogenesis of hBMSCs by targeting Wnt10 and eventually induces the inactivation of Wnt/β-catenin signalling. Subsequently, a microRNA-378 inhibitor-based therapy in an established mouse bone fracture model indicated that microRNA-378 inhibitor promotes bone formation and stimulates healing effects in vivo.

#### 5.1.2. MicroRNAs Involved in TGF-β/BMP/Smad Signalling

TGF-β/BMPs signalling pathways have widely recognized functions in bone formation and development. Disruptions of TGF-β/BMPs signalling have been implicated in various bone defects and diseases, such as osteoarthritis [116]. MicroRNA-92a was reported as a positive regulator to promote osteogenic differentiation of BMSCs by binding to the 3’ UTR of Smad6 [83]. Li et al. found that overexpression of microRNA-10b stimulates osteogenesis and inhibits adipogenesis of human adipose-derived stromal cells (hADSCs) in vitro [84]. Further pathway analysis indicates that microRNA-10b promotes osteogenic differentiation and bone formation through repressing Smad2 and TGF-β signalling pathways [84].

MicroRNA-214 was significantly downregulated during osteogenesis in BMSCs. Studies have shown that microRNA-214 targets the 3’ UTR of BMP2 to inhibit BMP signalling, which can be rescued by LncRNA (long non-coding RNA) KCNQ1OT1 [93]. A further definitive study found that overexpressed microRNA-214 repressed ALP activity and expression of osteogenic related genes OCN and OPN, thereby inhibiting osteoblast differentiation in BMSCs [94]. Similarly, microRNA-27a directly targets BMP2 and thus suppresses BMP signalling and negatively regulates osteoblast differentiation [95].

A clinical mixture of enamel matrix, emdogain, has been applied to induce osteogenesis and biomineralization [117]. MC3T3.E1 cells treated with emdogain indicated that the microRNA-30 family was dramatically downregulated, and further investigations revealed that the microRNA-30 family functions as a suppressor of osteogenesis by targeting Smad1 [96]. In addition, microRNA-26a and microRNA-199a-3p also directly target Smad1 that contributes to supporting osteogenesis and bone formation [97,98]. 

#### 5.1.3. MicroRNAs Involved in Osteogenesis Transcription Factors

Tian et al. reported that microRNA-23a binds to RUNX2, a master gene of osteoblast differentiation, leading to decreased osteogenesis of BMSCs. Their findings also revealed that, interestingly, CXC chemokine ligand-13 (CXCL13) can interfere with the interaction between microRNA-23a and 3’ UTR of RUNX2 and mitigates the inhibitory effects of osteogenesis [100]. MicroRNA-23b was also reported as an endogenous attenuator of RUNX2 in BMSCs, and infection with Ad-RUNX2 (adenovirus carrying the entire coding sequence of RUNX2) effectively rescued the suppression of osteogenesis in microRNA-23b-overexpressing BMSCs [101]. Moreover, using mouse models, significant osteoporosis was observed after agomir-23b injection into the caudal vein, and conversely, overexpression of RUNX2 enforced by combined injection of Ad-RUNX2 was found to soothe the bone defects induced by miR-23b [101]. Furthermore, several microRNAs, including microRNA 217, microRNA-10, microRNA-503-5p, and microRNA-505, have also shown negative regulatory effects in osteogenic differentiation by targeting RUNX2 [102,103,104,105].

In addition, upregulated microRNA-96 was identified in aged humans and mice with osteoporosis. At the molecular level, microRNA-96 regulated bone loss by targeting the osteogenic transcription factor OSX and inhibition of microRNA-96 in aged mice attenuated age-related bone loss [106]. Similarly, microRNA-27a-3p was upregulated in the serum of patients with osteoporosis. Further experiments have proven that microRNA-27a-3p functionally represses the osteoblast differentiation by directly binding to OSX, which synergistically promotes osteogenesis and bone formation [107].

### 5.2. MicroRNAs Involved in Adipogenesis and Fat Formation

#### 5.2.1. MicroRNAs Involved in Adipogenesis-Related Key Signalling Pathways

Some previous studies have shown that some microRNAs can regulate the two key signal pathways (namely Wnt signalling and TGF-β signalling pathways) that regulate adipogenesis. MicroRNA-9-5p was highly expressed in patients with osteoporosis [109]. Subsequent studies suggested that overexpression of microRNA-9-5p reduces osteogenesis and promotes adipogenesis by directly targeting the 3’ UTR of Wnt3a, a Wnt ligand that activates Wnt/β-catenin signalling [109]. Furthermore, 3T3-L1 cells with overexpressed microRNA-210 were led to cell enlargement with lipid droplets, whereas cells with microRNA-210 inhibition resulted in diminished adipogenic differentiation [111]. T-cell-specific transcription factor 7-like 2 (TCF7L2), the LEF/TCF family member responsible for triggering the downstream responsive genes of Wnt signalling, was identified as the direct target of microRNA-210 [111].

The role of microRNA-21 in adipogenesis of hADSCs has been investigated by Kim et al. [112]. They noticed that microRNA-21 positively regulates adipogenesis of hADSCs by directly binding to TGF-β1, which is known as an inhibitor for marrow fat formation in in vitro and in vivo models [112,118]. Further studies conducted by Kang et al. found that 3T3-L1 cells transfected with microRNA-21 presented higher levels of adipogenesis, which was indicated by higher expression levels of adiponectin [119]. In addition, overexpression of microRNA-199a-5p has been found to underlie increased adipogenic differentiation of BMSCs derived from aplastic anemia patients [120]. Further investigation demonstrated that microRNA-199a-5p promotes adipogenesis of hBMSCs in vitro by directly targeting transforming growth factor-beta-induced protein (TGFBI) [120], a multifunctional protein found in the extracellular matrix implicated in a variety of physiological processes.

#### 5.2.2. MicroRNAs Involved in Adipogenic Transcription Factors

Numerous studies have shown that the most frequently identified microRNAs in BMSCs perform as negative regulators of osteogenesis and favour adipogenesis. Zhao et al. have found that BMSCs isolated from aplastic anemia patients were induced to form adipocytes cells at the expense of osteoblasts [99]. The authors further identified that microRNA-204 is a critical regulator in aplastic anemia BMSC differentiation by targeting the 3’ UTR of RUNX2 mRNA [99]. Additionally, adipogenesis was enhanced, and osteogenesis was repressed, in microRNA-204-overexpressed cell models, while cell models that lost microRNA-204 functions displayed contrary effects [99]. Additionally, microRNA-637 was demonstrated to play a role in maintaining the balance between bone and fat formation by targeting OSX mRNA [108]. While the inhibition of OSX was found to positively regulate adipogenesis with the elevated expression levels of C/EBPα and PPARγ [108], microRNA-637 was demonstrated to significantly enhance de novo adipogenesis in nude mice [108].

MicroRNAs can directly control adipogenesis mainly through two principal transcription factors, C/EBPα and PPARγ [56,57,58]. MicroRNA-27a level was negatively correlated with PPARγ expression, and subsequent studies suggested that PPARγ is a direct target for microRNA-27a [85]. Moreover, data has shown that downregulated microRNA-27a promotes adipogenesis in BMSCs, and conversely, upregulated microRNA-27a attenuates adipogenesis but enhances osteogenesis in BMSCs [85]. Additionally, overexpressed microRNA-130a and microRNA-27b was shown respectively to strengthen osteogenesis and weaken adipogenesis in BMSCs by targeting PPARγ [86,87]. Sun et al. found that microRNA-31 binds to C/EBPα at both transcriptional and translational levels, leading to negative regulation of adipogenesis [113].

Furthermore, recent work has described that some microRNAs regulate adipocyte differentiation by interacting with factors that directly or indirectly affect the activities of PPARγ. For example, microRNA-146b was found as a positive regulator of adipogenesis. The negative correlation between microRNA-146b expression and SIRT1 was found in adipose tissue of obese mice [110]. In addition, it has been reported that microRNA-146b induces marrow fat formation by targeting SIRTI, which is known as an inhibitor for PPARγ [110].

Taken together, microRNAs discussed above that are involved in bone and fat regulation through regulating key signalling pathways and transcription factors are summarized in Figure 4. A complex microRNA regulation network involved in bone biological processes, including osteogenesis and adipogenesis, has been established [64,65,121].

## 6. MicroRNAs as Biomarkers and Therapeutic Targets

In the current clinical setting, the evaluation technologies for bone fragility and bone mineral density (BMD) are dual X-ray absorptiometry (DXA) and the fracture risk assessment tool (FRAX) [122]. However, these established tools have limitations. For example, the images of DXA are 2D instead of 3D, and bone microstructure, which is significant for bone strength, cannot be measured. Meanwhile, since the values of FRAX depend on BMD values, patients at high fracture risks but without osteoporotic symptoms of BMD are not suitable for FRAX. Therefore, it is important to explore more stable and suitable biomarkers to identify patients with a high risk of bone fracture usually presenting with bone formation defects and increased bone marrow adiposity.

MicroRNA investigations are increasing, which expands the concepts regarding biomarkers and therapeutic targets for bone defects. Specifically, the characterization of expressed and secreted microRNAs in the bone formation process might facilitate the process of selecting patients with high risks of bone diseases for targeted therapies. Such selection and identification of specific microRNAs correlated with bone diseases could be achieved by simple diagnostic tests, such as RT-PCR and routine blood tests. These diagnostic tests are more likely to obtain the general picture of differentially expressed microRNAs whose functions are related to different stages of bone pathology. Studies to date have provided some registered clinical studies of microRNA biomarkers, including phase 4 trials that observed potential microRNAs as biomarkers for disease progression in patients [123]. Furthermore, microRNA-based biomarkers could be combined with other biomarkers to improve the specificity and sensitivity of disease detection [124].

Some preclinical studies have shown that microRNAs with functional roles in inducing bone defects and bone/fat switch may become significant therapeutic targets. For example, a polyphenolic phytoestrogen with osteogenic properties called resveratrol could prevent the reduced bone formation and marrow adiposity in ovariectomized rats by inhibiting the expression of microRNA-338-3p but promoting the expression of RUNX2 [125,126]. Saferding et al. identified dysfunction of microRNA-146a paralleled osteoblast generation and bone formation, which was accompanied by reduced marrow adiposity [127]. The microRNA-146a deficient mice were protected from ovariectomy-induced bone loss and marrow adiposity [127]. Recently, the correlation between differentially expressed microRNAs and adipose tissue development and metabolism has been found in obesity and diabetes [128,129]. Brovkina et al. measured the mRNA and microRNA expression differences between obese individuals with (T2D+) and without type 2 diabetes (T2D−) [130]. Five upregulated microRNAs (microRNA-204-5p, microRNA125b-5p, microRNA-125a-5p, microRNA-320a, and microRNA-99b) were identified in T2D+ patients with obesity, while microRNA-23b-3p and microRNA-197-3p were increased in T2D− patients with obesity [130]. These miRNAs are involved in regulating matrix metalloproteinases and TGF-β signalling pathway to modulate adipogenesis [130]. Moreover, using the aptamer delivery system, BMSC-specific suppression of microRNA-188 resulted in enhanced bone formation and decreased marrow fat accumulation in aged mice [131]. It was found that microRNA-188 is a key regulator of age-associated bone/fat switch and might represent the potential therapeutic target for age-related bone/fat switch [131]. Additionally, bone growth defects and marrow adiposity are common skeletal complications for cancer patients undergoing irradiation and chemotherapy [3,132]. However, limited literature exists describing the microRNA controls in bone/fat switch following cancer treatment.

Currently, microRNA antagomir/inhibitor is the most common approach in in vivo studies when designing and exploring/establishing microRNA-based regulatory networks and therapeutic approaches [133].

## 7. Conclusions and Further Perspectives

Human and animal studies on the relationship between microRNAs and bone/fat homeostasis have provided valuable insights into microRNA regulation in bone and fat formation. Over the years, bone tissues, cell line models, patients with diverse skeletal conditions, and population-based work have been consecutively updating the network and library of microRNAs involved in regulating bone metabolism. However, the clinical translation of microRNAs in anti-bone/fat switch therapy is in its infancy. Numerous microRNAs still require further characterization and validation before some can be identified and selected for developing microRNA-based therapies.

Compelling future steps required include improving the comprehensive understanding of the emerging regulatory networks controlled by individual microRNAs and the roles of microRNAs responsive to osteogenic and adipogenic induction signals. Additionally, investigation into how the microRNA networks operate and how microRNAs integrate with other regulatory circuits is required. The challenge now is to identify the responsible microRNAs that are associated with bone defects and bone/fat switch through regulating commitment of BMSCs into bone or fat lineages, the identification of which will lay foundations for future further exploration and clinical translations into applications and modification of therapeutic genetic engineering.

## Figures and Tables

**Figure 1 ijms-22-07210-f001:**
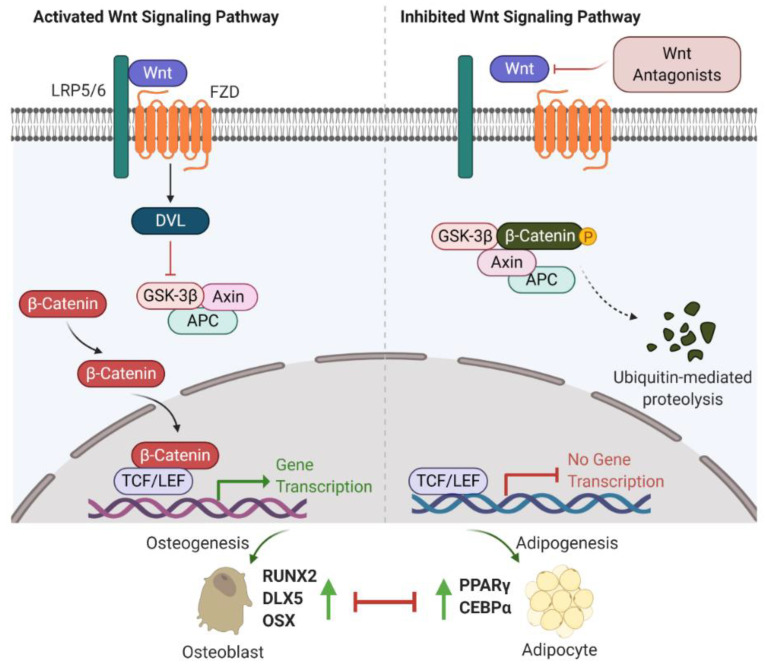
Wnt/β-catenin signalling involved in bone or fat regulation. (**Left**) In case Wnt ligand is binding to co-receptor complex, β-catenin is released from the destruction complex composed of GSK-3β, Axin and APC. This allows β-catenin to be accumulated and translocated into nucleus. By interacting with TCF/LEF transcription factors, β-catenin activates the osteogenic-related gene transcription program and promotes osteogenesis and bone formation. (**Right**) In case Wnt ligand is not able to bind to its co-receptor complex (e.g., in the presence of Wnt antagonists (sFRP-1, WIF-1, DKKs and sclerostin)), β-catenin is sequestered by the degradation complex, phosphorylated and subsequently degraded. Attenuated β-catenin enhances adipogenesis and marrow fat formation. Wnt: Wingless; LRP5/6: low-density lipoprotein receptor-related protein 5/6; Frizzled: FZD; DVL: disheveled; GSK-3β: glycogen synthase kinase 3β; APC: adenomatous polyposis coli; TCF/LEF: T-cell factor/lymphoid enhancer-binding factor; RUNX2: runt-related transcription factor 2; DLX5: distal-less homeobox 5; OSX: osterix; PPARγ: peroxisome proliferator-activated receptor γ; C/EBPα: CCAAT/enhancer-binding protein α.

**Figure 2 ijms-22-07210-f002:**
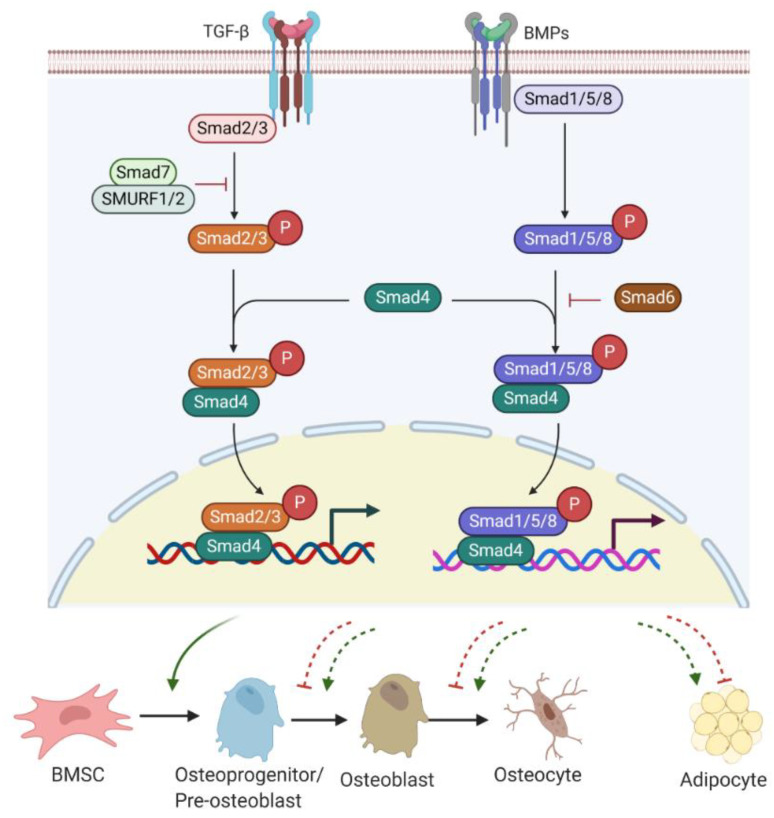
TGF-β/BMP signalling involved in bone or fat regulation. Active TGF-β or BMP binds to TGF-βR I/II or BMPR I/II receptor complex and can induce Smad-dependent signalling. R-Smads (Smad2/3 for TGF-β signalling and Smad1/5/8 for BMPs signalling) complexes with C-Smad, Smad4 and together translocate into the nucleus, where they regulate target gene expression. Smad7 with SMURF 1/2 negatively regulates Smad-dependent signalling by preventing Smad2/3 phosphorylation. Also, Smad6 inhibits the R-Smads/Smad4 complex to disrupt BMP signalling. TGF-β/Smad signalling promotes early differentiation of osteoprogenitors while it represses osteoblast maturation, mineralization, and transition into osteocyte. BMP-Smad signalling promotes almost each step during osteoblast differentiation and induces PPARγ expression for adipogenesis. However, TGF-β/Smad signalling negatively regulates adipogenesis and marrow fat formation. TGF-β: transforming growth factor β; BMPs: bone morphogenetic proteins; TGF-βR: TGF-β receptor; BMPR: BMP receptor; R-Smads; receptor-Smads; C-Smads: common-Smads; I-Smads: inhibitory-Smads; SMURF 1/2: Smad ubiquitin regulatory factor 1 and 2.

**Figure 3 ijms-22-07210-f003:**
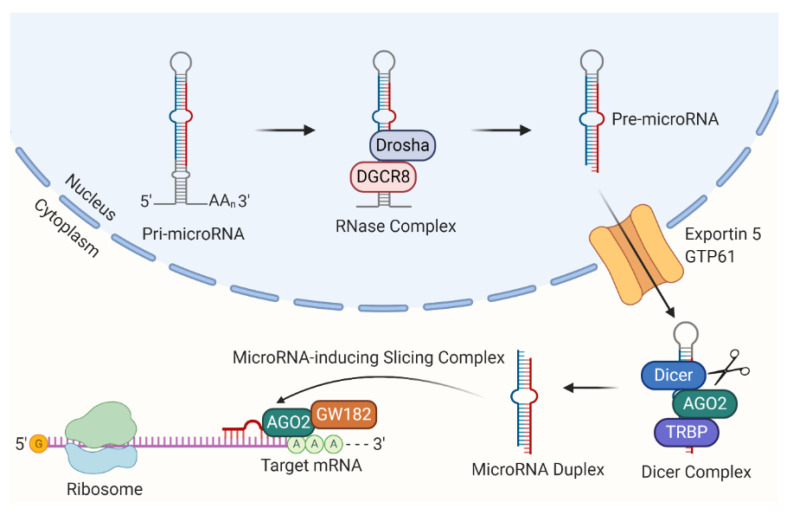
MicroRNA biogenesis. MicroRNAs are transcribed in the nucleus as pri-microRNAs. Pri-microRNAs are trimmed by Draosha-DGCR8 RNase complex and proceeded into pre-microRNAs that can be exported to the cytoplasm depending on the exportin 5/GTP61 translocation system. The Dicer complex binds to pre-microRNAs, which can be cleaved into microRNA duplexes containing guide and passenger strands. The guide strand often combines with AGO2 and GW182 as microRNA-inducing slicing complex and then interacts with the target mRNA, leading to target translation suppression or degradation. AGO2: argonaute 2; TRBP: transactivation response element RNA-binding protein.

**Figure 4 ijms-22-07210-f004:**
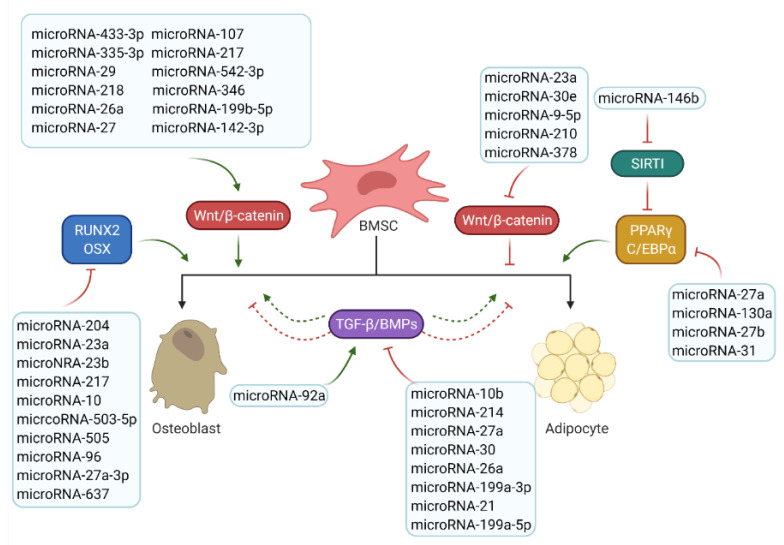
Roles of microRNAs in osteogenesis and adipogenesis. The list of reported microRNAs operates the network of key signalling pathways and factors to regulate bone and marrow fat formation. Green arrows mean positive effects and red arrow represent negative effects.

**Table 1 ijms-22-07210-t001:** A summary of microRNAs and their targets involved in osteogenesis and adipogenesis.

MicroRNA	Target	Reference
	Pro-osteogenesis	
MicroRNA-433-3p	Dkk1	[70]
MicroRNA-107	Dkk1	[71]
MicroRNA-335-3p	Dkk1	[72]
MicroRNA-217	DKK1	[73]
MicroRNA-29	DKK1, sFRP-2	[74]
MicroRNA-542-3p	sFRP-1	[75]
MicroRNA-218	DKK2, sFRP-2	[76]
	sclerostin	[77]
MicroRNA-346	GSK-3β	[78]
MicroRNA-26a	GSK-3β	[79]
MicroRNA-199b-5p	GSK-3β	[80]
MicroRNA-27	APC	[81]
MicroRNA-142-3p	APC	[82]
MicroRNA-92a	Smad6	[83]
MicroRNA-10b	Smad2	[84]
MicroRNA-27a	PPARγ	[85]
MicroRNA-130a	PPARγ	[86,87,88]
MicroRNA-27b	PPARγ	[86,87]
	Anti-osteogenesis	
MicroRNA-23a	LRP-5	[89]
MicroRNA-30e	LRP-6	[90]
MicroRNA-127 MicroRNA-136	N/A	[91]
MicroRNA-378	Wnt10a	[92]
MicroRNA-214	BMP2	[93,94]
MicroRNA-27a	BMP2	[95]
MicroRNA-30	Smad1	[96]
MicroRNA-26a	Smad1	[97]
MicroRNA-199a-3p	Smad1	[98]
MicroRNA-204	RUNX2	[99]
MicroRNA-23a & b	RUNX2	[100,101]
MicroRNA-217	RUNX2	[102]
MicroRNA-10	RUNX2	[103]
MicroRNA-503-5p	RUNX2	[104]
MicroRNA-505	RUNX2	[105]
MicroRNA-96	OSX	[106]
MicroRNA-27a-3p	OSX	[107]
MicroRNA-637	OSX	[108]
MicroRNA-9-5p	Wnt3a	[109]
	Pro-adipogenesis	
MicroRNA-146b	SIRTI	[110]
MicroRNA-204	RUNX2	[99]
MicroRNA-637	OSX	[108]
MicroRNA-9-5p	Wnt3a	[109]
MicroRNA-210	TCF7L2	[111]
MicroRNA-21	TGF-β1	[112]
MicroRNA-199a-5p	TGFI	[92]
	Anti-adipogenesis	
MicroRNA-10b	Smad2	[84]
MicroRNA-27a	PPARγ	[85]
MicroRNA-130a	PPARγ	[86,87,88]
MicroRNA-27b	PPARγ	[86,87]
MicroRNA-31	C/EBPα	[113]

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
