# Peer review of "Roles of MicroRNAs in Osteogenesis or Adipogenesis Differentiation of Bone Marrow Stromal Progenitor Cells"

_ijms, 2021, doi:10.3390/ijms22137210_

Round 1

Reviewer 1 Report

This manuscript is well organized and written with extensive review of BMSC osteogenesis and adipogenesis,  and further summarized the interaction of microRNAs with bone homeostasis. Just some minor comments for authors' consideration:

  1. Authors described BMPs as one family with pro-osteogenic functions. Please note that BMP family members may act differently. i.e. BMP3 is an antagonist for some other BMP members.
  2. It would be more appreciated if the cell-specific production and regulation of microRNAs were discussed. In terms of the effects on BMSC differentiation, microRNA functional mechanisms are through autocrine, paracrine or endocrine?
  3. Authors briefly mentioned a few clinical disease models. If applicable, a systematic discussion about microRNAs in models with bone loss and bone marrow adipose accumulation would be great, such as caloric restriction, diabetes, aging, irradiation, TZD treatment, ovariectomy etc.
  4. Lastly, line 356 states "negatively", however, seems this whole paragraph discussed "positive" effects of microRNAs.

Author Response

We highly appreciated the expert reviewer’ insightful and constructive comments and suggestions for this work. The manuscript has been revised based on the comments. Our responses for your specific comments are as follows:

Comment 1:

Thank you for your suggestion. We have now revised the description of BMP family in the revised manuscript line 144-148.

Comment 2:

We appreciated your advice. We have briefly discussed the microRNA functional mechanisms on bone/fat formation in revised manuscript line 313-321.

Comment 3:

The general discussion about microRNAs involved in bone/fat switch in differential disease models has been provided in revised manuscript line 571-594.

Comment 4:

Thank you for your comments. The “negatively” was previously used to indicate the function of GSK-3β and APC. We have now corrected this in revised the manuscript in line 369.

Reviewer 2 Report

This is a review of the role of miRNA on the balance between osteogenic and  adipogenic differentiation of BM stroll cells. The authors provide concise and concrete review on this subject.

Author Response

We appreciated your positive comments. Thank you.